# Quantum Neurobiology

**Melanie Swan [1],*, Renato P. dos Santos [2] and Franke Witte [3]**

[1]   Computer Science, University College London, London WC1E 6BT, UK
[2]   Physics, Lutheran University of Brazil, Canoas 92425-900, RS, Brazil; renatopsantos@ulbra.edu.br
[3]   Economics, University College London, London WC1E 6BT, UK; f.witte@ucl.ac.uk
*   Correspondence: melanie@blockchainstudies.org; Tel.: +44-20-7679-2000

**Abstract:** Quantum neurobiology is concerned with potential quantum effects operating in the brain and the application of quantum information science to neuroscience problems, the latter of which is the main focus of the current paper. The human brain is fundamentally a multiscalar problem, with complex behavior spanning nine orders of magnitude-scale tiers from the atomic and cellular level to brain networks and the central nervous system. In this review, we discuss a new generation of bio-inspired quantum technologies in the emerging field of quantum neurobiology and present a novel physics-inspired theory of neural signaling (AdS/Brain (anti-de Sitter space)). Three tiers of quantum information science-directed neurobiology applications can be identified. First are those that interpret empirical data from neural imaging modalities (EEG, MRI, CT, PET scans), protein folding, and genomics with wavefunctions and quantum machine learning. Second are those that develop neural dynamics as a broad approach to quantum neurobiology, consisting of superpositioned data modeling evaluated with quantum probability, neural field theories, filamentary signaling, and quantum nanoscience. Third is neuroscience physics interpretations of foundational physics findings in the context of neurobiology. The benefit of this work is the possibility of an improved understanding of the resolution of neuropathologies such as Alzheimer's disease.

**Keywords:** quantum biology; quantum neurobiology; quantum neuroscience; biological physics; neuroscience physics; quantum information science; neuropathology; theoretical neuroscience

## 1. Introduction

Quantum neurobiology is a topic within the broader field of quantum biology. The traditional concerns of quantum biology are studying quantum effects in biological systems such as magneto-navigation, photosynthesis, and energy transfer [1]. With the advent of quantum information science, the quantum biology research agenda is now being expanded to include also quantum information science approaches to biological questions [2], arguing that quantum models are needed to address the complexity of biology [3]. Representative projects include investigating excitation transport in photosynthetic light-harvesting complexes that indicate speedups analogous to those found in quantum algorithms [4] and explaining vibrational and environmental-assisted energy transport with quantum walks [5]. The word "quantum" refers to the scale of atoms and molecules ($10^{-9}$ to $10^{-15}$), namely atoms at the nanometer scale ($10^{-9}$), ions and photons at the picometer scale ($10^{-12}$), and sub-atomic particles at the femtometer scale ($10^{-15}$).

Quantum neurobiology has a parallel definition, with the first emphasis being the investigation of potential quantum effects in the brain [6], also including quantum information science methods being applied to neurobiological problems. This paper is principally concerned with the latter. Regarding potential quantum effects in the brain, on the one hand, there are proposals in favor of what might be termed the "quantum consciousness hypothesis" [7–9]. On the other hand, many scientists are careful to distinguish that they do not endorse this idea, instead supporting the possibility that the mathematical structure

of quantum mechanics may help to explain neural behavior, but not the conjecture that there is something quantum-like taking place in the brain [10–12].

Other research programs use quantum information science methods to model cognitive processes such as perception, memory, and decision-making, without taking a view regarding whether quantum effects operate in the brain [13]. Research programs also target less-contentious topics of quantum-related activity in the brain through quantum events [14] and superdeterminism [15]. Superdeterminism interprets quantum mechanics as an effective statistical theory of hidden variables as opposed to one of fundamental indeterminism. Evolutionary reasons might explain why biological systems of sufficient complexity display quantum-like behavior, independent of the physical origin of quantum phenomena in physics. In this vein, quantum biology might crosspollinate back to foundational physics with hidden variables formulations as a practical method for operating in quantum domains (e.g., hidden-variable models of Bell correlations [16] and Kolmogorov-related probability formulations [17]). In any case, the first-principles step would seem to be the enumeration of the underlying physiological processes as the building blocks that might then be examined in relation to higher-order cognitive behavior [18].

Quantum neurobiology extends classical neurobiology as a multidisciplinary field relating biology to the fundamental and clinical neurosciences, investigating the form and function of neurons, glia, axons, and dendrites in the nervous system, individually and in ensemble, in health and disease. Various approaches are used to study multiscalar behavior in the areas of neural signaling transduction and transmission, neural circuits and systems neurobiology, nervous system development and aging, and the neurobiology of disease and intervention, including by examining how quantum properties enhance cellular function, with medical implications for neuropathology diagnosis and treatment [19].

### 1.1. The Human Brain

The brain is among the most complex systems known [20], with a behavior spanning nine orders of magnitude-scale tiers in ways that have yet to be fully elucidated (Table 1). However, in the "big data" era, the requirements associated with modeling the brain (which has an estimated 86 billion neurons and 242 trillion synapses [21]) are coming within reach. Biology became an information science with the advent of genomics, and neuroscience is one of the fastest-growing areas in information biology, with data acquisition outpacing that of most other biomedical fields [22]. Whole-brain scanning is revealing the simultaneous activity of hundreds and thousands of neurons [23] with large-scale cortical recordings [24] and whole-brain activity logging in behaving organisms [25]. Microscopy advances obtain a single-molecule resolution that captures synaptic proteins at dendritic spines, myelination along axons, and presynaptic densities at dopaminergic neurons with expansion light sheet microscopy [26]. Neuropathologies may be treated at the synaptic scale with novel stem cell therapies and pharmacological compounds to reverse the effect of dysfunctional genes [27].

**Table 1.** Levels of organization in the brain (Adapted from [28,29]).

| No. | Level | Size (Decimal) | Size (m) | Size (m) |
|-----|-------|----------------|----------|----------|
| 1 | Nervous system | 1 | >1 m | $10^0$ |
| 2 | Subsystem | 0.1 | 10 cm | $10^{-1}$ |
| 3 | Neural network | 0.01 | 1 cm | $10^{-2}$ |
| 4 | Microcircuit | 0.001 | 1 nm | $10^{-3}$ |
| 5 | Neuron | 0.000 1 | 100 μm | $10^{-4}$ |
| 6 | Dendritic arbor | 0.000 01 | 10 μm | $10^{-5}$ |
| 7 | Synapse | 0.000 001 | 1 μm | $10^{-6}$ |
| 8 | Signaling pathway | 0.000 000 001 | 1 nm | $10^{-9}$ |
| 9 | Ion channel | 0.000 000 000 001 | 1 pm | $10^{-12}$ |

In high-throughput connectomics, ongoing work from the Allen Institute demonstrates terabyte-scale processing for contemporary neuron reconstruction [30] and petabyte-scale

next-generation dataset acquisition methods [31]. To complete the human connectome (and even the mouse connectome [32]), a qualitatively different form of computation may be required, similar to the technology-driven inflection point in the sequencing of the human genome, enabling its completion in 2001 [33]. Human connectomes are not an immediate prospect, as the whole-brain fruit fly connectome was only completed in 2018 [34]. The imaging, data processing, and storage requirements may be 1 zettabyte per human connectome [35], which compares to the 59 zettabytes of data generated worldwide in 2020 [36]. Neurobiological informatics data include not only genomics and connectomics, but also synaptomics (mapping of synapses across the brain) and synaptosomics (the synapse proteome, 1000 proteins implicated in 130 brain diseases) [37].

### 1.2. Quantum Neurobiology

These state-of-the-art advances in neurobiology pave the way for quantum neurobiology and facilitate the research aim of whole-brain neuroscience: full-volume, three-dimensional analysis of the entire brain at multiple spatial and temporal scales. An immediate practical task is integrating data obtained simultaneously from EEG, MEG, fMRI, and diffusion tractography (nerve tract data) [38]. Quantum approaches are needed as supercomputing (only able to model one third of the human brain in a recent project [39]) and other classically based methods make it clear that new platforms are needed for the next phases of neuroscience data analysis. Simultaneously, quantum information science is emerging as a vastly more scalable platform with three-dimensional modeling capabilities appropriate to the representation of real-life brain phenomena such as neurons, glia, and dendritic arbors. Quantum approaches allow new classes of neurobiological problems to be addressed more fully, such as the investigation of neural signaling with synaptic integration (aggregating thousands of incoming spikes from dendrites and other neurons) and electrical–chemical signal transduction (incorporating neuron–glia interactions at the molecular scale). This work describes the three areas of activity developing in quantum neurobiology (Table 2) and proposes a novel theory of neural signaling (AdS/Brain, based on the AdS/CFT correspondence (anti-de Sitter space/conformal field theory)).

**Table 2.** Quantum neurobiology: three areas of quantum information science study.

| 1. Waves, Protein Folding, Genomics | 2. Neural Dynamics | 3. Neuroscience Physics |
| --- | --- | --- |
| Waves | Superpositioned Data | AdS/Neuroscience |
| ▪Quantum EEG | Quantum Probability | ▪AdS/Brain |
| ▪Quantum MRI | ▪Updating (QBism) | ▪AdS/Memory |
| Quantum Protein Folding | Neural Field Theories | ▪AdS/Superconducting |
| Quantum Genomics | ▪Synchrony | ▪AdS/Energy |
| ▪Sequencing | Filamentary Dynamics | Neuronal Gauge Theories |
| ▪Gene Expression | Quantum Nanoscience | Network Neuroscience |
| ▪Secure Transmission | ▪Nanoparticle Fab | Random Tensors |
| Quantum SNNs | ▪Molecular Codes | ▪Melonic Diagrams |

Symbol (▪) means the sub-levels.

## 2. Waves, Protein Folding, and Genomics

### 2.1. Wavefunctions: EEG, fMRI, CT, PET Integration

The first widespread class of quantum neurobiology applications is the interpretation of empirical data from various neural scanning modalities with wavefunctions and quantum machine learning. The EEG-detectable potentials given off by the scalp have been analyzed since 1875 [40], but a fuller picture of neural signaling also includes waveforms related to astrocyte calcium signaling, neurotransmitter activity, and dendritic spikes [41]. Although quantum mechanical wavefunctions are naturally suggested, the intractability of the Schrödinger wave equation has traditionally meant that EEG data are interpreted with effective nonlinear wave models [42]. Quantum algorithms are now supplanting this effort and being used to reconstruct medical images from MRI, CT, and PET scanners [43].

Near-term applications could be in the area of quantum BCIs (brain–computer interfaces), interpreting EEG waveform data in a brain–machine communications network [44,45].

## 2.2. Quantum EEG

Quantum machine learning is emerging as an indispensable technique for finding the best wavefunction to fit the copious amounts of EEG data generated. A typical problem is classifying EEG data for Parkinson's disease patients as potential candidates for Deep Brain Stimulation, analyzing 794 features from each of 21 EEG channels [46]. Quantum machine learning is the application of machine learning techniques in a quantum environment, formulating classical data with quantum methods, and also studying quantum problems with machine learning methods [47]. Quantum formulations are available for the three main machine learning architectures: neural networks [48], tensor networks [49], and kernel learning [50]. A quantum perceptron (core machine learning unit) has been developed for available quantum processors (the IBM Q-5 Tenerife) [51]. Quantum neural networks have been proposed for EEG wavefunction modeling, in a standard gate-model quantum circuit layout using quantum convolutional neural networks (CNN) [52], and also in a more sequentially oriented quantum recurrent neural network (RNN) [53].

An alternative to quantum machine learning is quantum spike-activated neural networks (SNNs)—a bio-inspired neuromorphic computation model with threshold-triggered activation similar to the natural neural firing of the brain [54]. Exemplar quantum SNN projects use Josephson junctions to study emergent behavior [55] and accelerated matrix processing via synaptic weighting [56] and superposition modeling [57].

Framed as a signal processing problem, EEG data interpretation is an exercise of noise filtering followed by feature extraction and classification. EEG data used in BCIs, for example, have a low signal-to-noise ratio due to noise. A quantum approach applies filtering algorithms based on advances in processing techniques (Kullback–Leibler spatial patterns and Bayesian learning) [44] in a quantum recurrent neural network (QRNN) format. The QRNN characterizes a nonstationary stochastic signal as time-varying wave packets, interpreted with the Schrödinger wave equation and a Hamiltonian (energy operator). The QRNN outperforms traditional Kalman filtering methods and is tested on real-time EEG data and BCI competition test data. The feature extraction and classification portion of EEG data analysis is likewise performed with various quantum machine learning methods such as entropy-based quantum support vector machines [58], quantum-inspired evolutionary algorithms [59], and independent component analysis, wavelet transforms, and Fourier transforms [60]. Finally, quantum methods are facilitating a new level of data resolution in the examination of EEG data. One project investigates single-trial event-related potentials (EEG segments time-locked to a cognitive events) with universal cortical building blocks in the time and frequency domains [61], and another models electrical signals and calcium-ion interactions together in a path integral approach [62].

## 2.3. Quantum MRI (Radiology)

Aside from EEG, MRI scans are the other main imaging modality for neural data. Quantum machine learning techniques could also prove central to MRI data interpretation in the classification of 120 different types of brain tumors, as classical deep learning networks are already used to identify the six most common tumor types (glioma, brain metastases, meningioma, adenoma, and neuroma) [63]. Qutrits (three-level quantum states) may be conducive to brain tumor analysis since many quantum states are not binary. One proposal suggests that a qutrit model might better correspond to grayscale imaging data, using a quantum neural network model to segment brain lesions [64]. Whereas qubits are a relatively simple system, expanding to higher dimension qudits (quantum information digits) is non-trivial as it is difficult to quantify the quantum correlations in the system (using the diagonalization of correlation matrices for bipartite systems) [65]. Other quantum MRI tumor segmentation projects use a quantum entropy classification method [66] and a quantum filtering technique (for noise reduction preprocessing) together with a

quantum artificial immune system-inspired SoftMax function in a deep spiking neural network (SNN) architecture [67]. Quantum algorithms are also deployed to analyze CT scans; for example, to classify quantum data comparing COVID-19 and non-COVID-19 patient influenza and virus pneumonia lung CT scans, analyzed with TensorFlow Quantum and a D-Wave Systems quantum annealer [68].

## 2.4. Quantum Protein Folding

Protein folding is an NP-hard computationally complex problem with advances in both classical and quantum methods. The challenge is to predict the three-dimensional structure that a protein will adopt based on the underlying sequence of amino acids. Many neurodegenerative diseases (such as Alzheimer's and Parkinson's) are thought to be caused by an accumulation of misfolded proteins [69]. Classically, an important project is AlphaFold, as Google's DeepMind team extends its success in game playing [70] to protein folding, as shown in the CASP-14 data competition [71]. An attention-based mechanism is used to obtain atomically precise configurations by paying attention to global constraints such as available space as opposed to exclusively local sequence interactions.

Quantum methods are also progressing, particularly with quantum annealing machines that model protein folding as a low-energy optimization. A lattice is used to represent the spatial location of the different amino acid sequences in the protein. Although annealers can easily analyze the median length of a human protein (375 amino acids), research often focuses on neuropeptides as short protein strings that can be readily employed as intervention targets. One lattice-based quantum protein folding project studied 30,000 protein sequences with protein Hamiltonians, finding that simple manipulations substantially improve folding performance [72]. A related project demonstrated the lattice-based folding of a 7-amino acid neuropeptide (with the IBMQ Poughkeepsie 20-qubit quantum computer) [73]. Quantum walks are an alternative to lattice structures, as the QFold project proposes a quantum algorithm based on the torsion angles of amino acids, deployed with quantum walks (on the IBMQ Casablanca quantum processor) [74].

## 2.5. Quantum Genomics

The quantum properties of DNA have been proposed for use in sequencing (for example, interpreting electron tunneling current–voltage differences between the four nucleotide bases as a strand of DNA passes through a nanopore [75]), but quantum methods are mainly deployed in sequence reconstruction (aligning and merging reads to reassemble the original genome). Quantum algorithms have been proposed (for both gate-array and quantum annealing machines) to accelerate DNA sequence reconstruction [76] and demonstrated on quantum annealing platforms to reconstruct short sequences (seven nucleotides) [77]. Quantum annealing machines are also used in basic research to assess the binding affinity of gene regulatory proteins to the genome [78]. In other quantum genomics demonstrations, a quantum machine learning algorithm is implemented for Alzheimer's disease to identify neurons that have irregular numbers of chromosomes (copy number variation) with a Hamming distance-like genomic quantum classifier (tested on the IBMQX2 and IBMQ 16 Melbourne quantum platforms) [79]. Another project proposes a cell culture analysis technique to assess the clonogenic survival potential of a cell to grow and form a colony based on a quantum information-theoretic classifier [80].

Quantum networks offer possibilities for the secure transfer of genome files; for example, in next-generation federated data sharing for large-scale research using encrypted blockchain-based quantum networks [81]. The benefits of blockchain technology as a secure smart network automation technology, particularly for genomic data sharing, have been proposed [82] and have seen practical deployment for genomic data privacy in whole-human genome sequencing projects such as Nebula Genomics [83]. Non-fungible genome tokens (NGTs) provide users with permanent ownership of their genomic data on a publicly readable ledger, enabling user-controlled, remunerated, transparent data-sharing [84]. Other research uses quantum networks for secure encrypted genome transmission, sending

data immediately as it is sequenced, demonstrating the world's first quantum cryptography transmission of whole-genome sequence data [85].

## 3. Neural Dynamics

The second area of application in quantum neurobiology is neural dynamics, which consists of superpositioned data modeling evaluated with quantum probability, neural field theories, filamentary dynamics, and quantum nanoscience.

### 3.1. Superpositioned Data and Quantum Probability

Studying complex systems often involves finding otherwise hidden correlations in datasets. Progress has been made with deep learning networks and also now in the modeling of classical data with quantum methods [86]. Superpositioned data are data modeled in superposition as the quantum information representation of all possible system states simultaneously. A standard model for superpositioned data is neural signaling, in which system elements ("neurons") exist simultaneously in two or more states until collapsed in a measurement ("firing event"). The basic setup is a two-state model in which elements exist in both inactive (quiescent) and active (firing) states until measured [87]. Another model is a neural field theory with a three-state neural signaling system in which the neuron states are quiescent, active, and recovering [88]. Neural signaling models are used, both as a general heuristic to represent any kind of multistate system, and in particular to study biological signaling in the brain.

Quantum information science approaches require a formulation of quantum probability (complex probability amplitudes) since the quantum properties of superposition and interference violate classical total probability [89]; for example, leading to conjunction and disjunction fallacies in commutativity [13]. Hence, quantum probability has been formulated by von Neumann and others to apply quantum mechanical rules to probability assignment [90,91]. A standard quantum information science primitive (building block) used as a quantum variant of total probability is obtained through POVMs (positive operator valued measures). POVMs are positive measures on a quantum subsystem of the effect of a measurement performed on the larger system and give an interference term for incompatible observables [92]. The main interpretation of quantum probability is with the Born rule (a solvable probabilistic formulation of quantum mechanics), but there can be others. Quantum Bayesianism methods, notably QBism ("cubism") [93], is an emerging standard for considering quantum system updating as the quantum version of Bayesian updating that includes observer-based (subjective) aspects [94].

### 3.2. Neural Field Theories

Neural field theories are a physics-based approach for modeling large-scale brain behavior. An empirical project involving neural dynamics and neural field theories is combining data from different neural scanning modalities (EEG, fMRI, CT, and PET scans) into a comprehensive view of brain activity. Integrating EEG and fMRI data, for example, entails multiple spatiotemporal scales and dynamics regimes [38]. A key finding is that epileptic seizure can be modeled by chaotic dynamics, which are understood, but the normal resting state of the human brain is more complicated and is perhaps explained by instability-bifurcation dynamics, in which there is one system organizing parameters, such as an orbit that is interrupted periodically by countersignals to trigger a neural signal [95]. A standard modeling element is the Hopf bifurcation—a system-critical point at which a periodic orbit appears or disappears due to a local change in stability [96,97].

Neural dynamics vary by scale and display unrecognized statistical distributions at the most complex tiers of collective behavior [38]. Different neural dynamics models are deployed respectively at the four main scales tiers of activity: single-neuron, local ensemble, population group, and whole-brain. Small populations, represented by neural ensemble models, often follow a normal statistical distribution, which can be described with a Gaussian and modeled with a regular linear Fokker–Planck equation; in contrast, if not normally

distributed, the distribution might be non-Gaussian, but still a known distribution such as a power law that can be modeled with specialized non-linear or fractional Fokker–Planck equations. However, larger scale populations of neurons and whole-brain analysis have unrecognized statistical distributions and require more sophisticated dynamics methods such as Wilson–Cowan mean field equations, the Jansen–Rit model, Floquet periodicity, and oscillatory analysis [38]. Large-scale neural dynamics is an active research area with a substantial opportunity for quantum neurobiology to make a contributive impact. Statistical approaches in neural field theories continue to contribute to the development of new mathematics for the understanding of theoretical neuroscience problems [98].

Biological systems have additional requirements compared with other domains in that organisms do not exist in isolation but are rather constantly interacting with the environment and changing their behavior as a result. Hence, feedback loops and updating are important in quantum biological system modeling. One such quantum information biology approach is based on two-state superposition data modeling, system self-measurement, and open system (environmental interaction) evolution dynamics [87]. A self-observation feedback loop is included as the quantum version of the Helmholtz sensation-perception theory (a unitary operator describes the process of interaction between the sensation and perception states). Open system evolution dynamics are provided by a Lindbladian quantum master equation. The model applies to all biological system scales (including protein, cell, brain, human behavior, and ecosystem) and is tested to study epigenetic evolution and the gene regulation of glucose-lactose metabolism in *E. coli* bacteria [99] and in a neural code mapping model of human decision-making states [100].

A final topic in neural dynamics is synchrony—a proposed bulk property of the brain. Synaptic signals arrive simultaneously but travel varying distances and therefore must have different propagation speeds [101]. Cortical recordings further reveal that spontaneous traveling waves are a general topological property of large-scale neural behavior [102]. One project studies synchrony in axon propagation speeds from data recorded at multiple spatial scales [103]. A general framework is proposed to integrate microscale current sources (produced by local field potentials at membrane surfaces) in a macro-columnar structure. Another project applies the Kuramoto model, a standard formulation for studying synchrony in nonlinear systems ranging from insect swarms to superconductors [104]. A solution is produced for the three main synchronization phenomena in Kuramoto networks (phase synchronization, chimera states, and traveling waves) with insight into complex behavior arising from connection patterns in nonlinear networked systems.

### 3.3. Neurofilamentary Dynamics

Neurofilaments are neuron-specific proteins that provide structural support to the neuronal cytoskeleton and are implicated in neural signaling (axonal and synaptic) via dynamical behavior. In synaptic signaling, neurofilamentary proteins are differentially expressed in the presynaptic and postsynaptic compartments of glutamatergic (excitatory) and GABAergic (inhibitory) synapses [105]. Neurofilamentary dynamics likewise have a role in axonal signaling. Research finds that the axon processes information at multiple time scales [106,107] in the shape of vortex-like signals that can be captured with quantum optics [108]. An axon has thousands of densely packed neurofilaments beneath the membrane. An intricate mechanism of electromagnetic and ionic signaling is suggested in that the electromagnetic resonance of neurofilaments first identifies relevant paths or circuits (branch selection) extremely rapidly at a microsecond speed ($10^{-6}$), which then serves as an input to the ion channel transmission that proceeds on the order of milliseconds ($10^{-3}$) [106]. Specifically, four ordered structures in the cytoskeletal filaments were shown to exchange energy approximately 250 microseconds before a neuron fires [107]. The research program integrates multiple time domains into a single temporal architecture, extending the traditional Hodgkin–Huxley model used to study neural signaling, branch selection, spike time-gap regulation, and synaptic plasticity [108]. Understanding filamen-

tary dynamics is important as these proteins are proposed as a blood-based biomarker of neurodegenerative pathology, overcoming some of the challenges of amyloid-beta and tau proteins as the traditional diagnostic markers for Alzheimer's disease [109]. For example, one study found blood-based neurofilamentary protein fragment levels to be eight times higher in neurological disease patients than controls [110].

### 3.4. Quantum Nanoscience for Neurobiology

Quantum nanoscience is the study of nanostructured systems that incorporate and exploit quantum effects [111]. The fabrication of integrated circuits and nanomedicine are two of the primary applications of quantum nanoscience [112,113]. Both are relevant to quantum neurobiology—nanomedicine most directly—and nanocircuits in the effort to create standardized quantum neural circuits to test behavior, pathological response, and pharmacological intervention. Various projects are attempting to identify the structural–functional organization of neural circuits per connectome project data, serial electron microscopy, trans-synaptic tracing, and single-cell transcriptomics [114]. In nanomedical fabrication, nanoparticles (precision-engineered objects with dimensions less than 100 nm) are the focus of quantum neurobiological modeling and drug design.

### 3.4.1. Nanoparticle Neuroscience

Nanoparticles allow therapeutics to be delivered across the blood–brain barrier (BBB) into the brain. A nanoparticle has a relatively large surface area and pores for housing therapeutic agents. Adjusting the size and molecular weight of the delivery system containing the drug can be used to target where the nanoparticles accumulate in the body and the tissue that can be accessed. One project creates a blood–brain barrier-crossing nanoparticle drug delivery platform to treat secondary injuries associated with traumatic brain injury that can lead to Alzheimer's, Parkinson's, and other neurodegenerative diseases [115]. The therapeutic is a small interfering RNA (siRNA) molecule designed to inhibit the expression of the tau protein (thought to play a role in neurodegeneration). The solution encapsulates therapeutic agents into biocompatible nanoparticles with precisely engineered surface properties to enable their transport into the brain and indicates a 50% reduction in the expression of the tau protein as a result. In addition to nanoparticles, other contemporary neuropathology resolutions are being explored such as CRISPR/Cas9 therapeutic strategies (supplying or blocking proteins) in pre-clinical Alzheimer's disease models [116].

### 3.4.2. Molecular Codes

A recent advance is precision molecular control, performed with quantum error correcting methods (molecular codes), highlighting the integration of quantum information-based systems and physical systems. Molecular codes are an extension of GKP bosonic codes (Gottesman, Kitaev, Preskill) [117]). Bosonic codes are a method of quantum error correction instantiating both the physical qubit and the logical qubits that protect it in a self-contained system (the continuous variable environment of a harmonic oscillator) [118]. GKP bosonic codes correct errors (seen as molecular displacement) by reorienting the position and momentum of an oscillator (oscillatory molecule) with known symmetric rotations. Molecular codes extend GKP bosonic codes by allowing rotations to be performed on asymmetric rigid bodies in free space, in quantum systems ranging from oscillators to diatomic and polyatomic molecules [119]. Error-corrected molecular control is an important capability in the quantum information-theoretic modeling of neural behavior as neural circuits are instantiated in quantum hardware.

### 3.4.3. Autonomous Robotic Nanofabrication

Quality-controlled nanomedical fabrication might proceed with autonomous agents. One project demonstrates the autonomous robotic nanofabrication of supramolecular structures from single molecules [120]. The method consists of controlling single molecules with the machine learning agent-based manipulation of scanning probe microscope actuators

(using reinforcement learning (goal-directed updating) to remove molecules autonomously with a scanning probe microscope from a supramolecular structure).

## 4. Neuroscience Physics

The third area of quantum neurobiology applications is neuroscience physics, which is the neuroscience interpretation of foundational physics findings. Applications discussed here include a suite of AdS/Neuroscience theories based on the AdS/CFT correspondence (AdS/Brain, AdS/Memory, AdS/Superconducting, and AdS/Energy (brain Hamiltonian)), neuronal gauge theories (symmetry-breaking, energy-entropy balances), network neuroscience, and random tensors (high-dimensional systems).

Of particular interest is neural signaling—a problem involving synaptic integration (aggregating thousands of incoming spikes from dendrites and other neurons) and electrical–chemical signal transduction (incorporating neuron-glia interactions at the molecular scale). The standard compartmental models used in computational neuroscience are not equipped with the multi-variable partial differential equation (PDE) functionality needed to model inter-neuronal spatial interactions [121,122]. For example, diffusion–reaction equations are one possibility for integrating the activity of dendritic spikes that involves astrocyte calcium signaling, protein cascades in dendritic arbors, and the proton and ion-based transfer of molecules [123], all of which take place on the quantum (atomic and subatomic) scale [124]. Proposed quantum neurobiological solutions to the modeling of neural signaling consider multiscalar models, phase transition, dynamical nonlinear systems, energy–entropy relations, and high-dimensional representation.

### 4.1. AdS/Brain

AdS/Brain is a multiscalar theory of neural signaling based on the AdS/CFT correspondence, which incorporates the four scale tiers of network, neuron, synapse, and molecule [125]. The theory is the first example of a multi-tier interpretation of the AdS/CFT correspondence with successive levels of bulk–boundary correspondence. The suggested implementation of the AdS/Brain theory is a matrix quantum mechanics formulation (multi-dimensional matrix model [126]) with bMERA (brain) random tensor networks evolved with Floquet periodicity-based neural dynamics.

The AdS/CFT correspondence (anti-de Sitter space/conformal field theory) is a theory positing that a physical system with a bulk volume can be described by a boundary theory in one less dimension [127]. Specifically, the theory (gauge/gravity holographic duality) states that a gravity theory (bulk volume) is equal to a gauge theory or quantum field theory (boundary surface) in one less dimension. The work constitutes one of the most-cited papers in any field (over 21,000 references as of December 2021), with applications in all physics arXiv areas [128].

The AdS/CFT correspondence offers two perspectives of the same system and the mathematics for solving in either direction. A typical bulk-to-boundary use case is the AdS/SYK (Sachdev-Ye-Kitaev) formulation, starting with a known classical gravity theory (Einstein gravity) in the bulk to solve for an unknown quantum field theory describing a superconducting material on the boundary. The mathematics of black holes (classical gravity bulk) and unconventional materials (boundary) can be linked in that the two systems have similar properties related to mass, temperature, and charge [129]. In the other direction, boundary-to-bulk deployments start with a known quantum field theory on the boundary and attempt to define a theory of emergent structure such as an unknown quantum gravity theory in the bulk [130]. Establishing bulk–boundary mappings, including a quantum error correction setup (protecting a logical qubit in the bulk by linking it to an ancilla of physical qubits in the boundary [131]), is an active research area [132].

The AdS/Brain theory proposes the first instance of a multi-tier correspondence (multiple graduated levels of bulk–boundary relationships) to instantiate the four scale tiers of brain network, neuron, synapse, and molecule (and could be expanded to other tiers). The model accommodates the entirety of the brain's neural signaling processes between

axon, presynaptic terminal, synaptic cleft, postsynaptic density, and dendritic spiking potentials from dendrite to soma. The bulk–boundary pair relationships are network–neuron, neuron, synapse, and synapse–ion. The scales and measured signals are local field potentials at the brain network level ($10^{-2}$ m), action potentials at the neuron level ($10^{-4}$ m), dendritic spikes at the synapse level ($10^{-6}$ m), and ion docking at the molecular level ($10^{-10}$ m).

The AdS/Brain theory addresses the renormalization requirement in multiscalar systems (the ability to view a physical system at different scales). Renormalization programs must tackle the infinities that arise in quantum mechanics to reflect the fact that all possible particle locations and events can actually occur. Various renormalization group (RG) methods have been proposed as a mathematical apparatus for smoothing systems so that they may be viewed at different scale tiers on the basis of different parameters (degrees of freedom). A key advance is the multiscale entanglement renormalization ansatz (MERA), a tensor network structure that implements an iterative coarse-graining scheme to renormalize quantum systems on the basis of entanglement or other attributes [133]. The MERA tensor network consists of alternating layers of disentanglers and isometries that consolidate a multi-tier system into a single view—a structure conducive to the AdS/Brain theory and applied in a bMERA (brain) implementation.

The second requirement the AdS/Brain theory addresses is the issue that different neural dynamics paradigms define the system evolution at each scale tier of the neural signaling operation [38]. Floquet periodicity [134,135] propelled with continuous-time quantum walks [136] is selected as the basis for a multiscalar model of brain network, neuron, synapse, and ion channel dynamics, as these formalisms flexibly accommodate varying dynamical regimes within a system.

### 4.2. AdS/Memory

AdS/Memory is a neuroscience application of the AdS/CFT correspondence that examines the problem of information storage. The research program applies the AdS/CFT correspondence (in the form of black hole physics) to the computational neuroscience problem of memory formation [137]. Black holes and brains are efficient at storing information, and critically excited states might be the basis. A quantum neural network with holographic properties (entropy scaling by area not volume) is introduced. The quantum optical neural network (with qudit-based bosonic modes) produces critical states (neuron excitatory synaptic connections based on gravity-like interaction energy) that have an exponentially enhanced capacity to store information. What is new is the investigation of what a system can do in a highly excited state (as opposed to exclusively finding the system's ground state and related energy tiers). The largest memory capacity quantum state might not necessarily be the ground state, but rather a highly excited critical state. There could be immediate implications for quantum memory and also in quantum neurobiology in substantiating the conceptualization of neural signaling as a criticality triggered phase transition. The consideration of system extremes is a method also being applied, for example, to find new matter phases in systems that do not reach thermal equilibrium [134].

### 4.3. AdS/Superconducting

AdS/Superconducting uses the AdS/CFT correspondence to study phase transition, which is not well understood in various domains including neural signaling and superconducting materials. Explaining how materials become superconducting at high critical temperatures could be practically useful in producing superconducting chips that do not require super-cooling. One approach deploys the AdS/CFT correspondence (in the AdS/SYK program of using a known gravity theory to find an unknown quantum field theory for a superconducting material) to study superconducting systems [138]. The setup is a toy-model black hole, constrained to a box (like the gas-in-a-box or particle-in-a-box model systems). The black-hole-in-a-box (unlike real black holes) can be manipulated such that a condensate halo (of some material) forms around it. When an external electri-

cal field is applied (like turning on a battery), the condensate becomes superconducting due to the Higgs mechanism. In the general case (confirmed by Large Hadron Collider experiments), the Higgs mechanism "gives particles their mass" as the Higgs field is a universal field throughout the universe that causes particles to become "heavy" as they pass through a medium, giving them drag, or mass [139]. In the black hole condensate situation, the particles that become massive are photons, preventing electric and magnetic fields from traveling through the medium, causing the medium to become superconducting (electrons flowing freely with infinite conductivity and zero resistance). The result is the AdS/Superconducting model, an experimental model for studying phase transition, particularly in systems with ordered-disordered phases such as neural signaling.

*4.4. AdS/Energy (Brain Hamiltonian)*

The AdS/Brain theory provides a generic multiscalar model of neural behavior interpretable at various bulk–boundary scale tiers with the AdS/CFT mathematics, which renormalize entanglement (correlations) across system levels. Although entanglement is the primary multiscalar quantity, energy-related formulations (expressed as a Hamiltonian) are also possible. A first law of thermodynamics (the first law of entanglement entropy (FLEE)) has been defined to posit that a change in boundary entropy is equivalent to a change in bulk energy (Hamiltonian) [140]. Energy formulations are central to quantum systems, but a formalism did not exist previously for solving the AdS/CFT correspondence in terms of energy. The implication is that the AdS/CFT correspondence is immediately connected to the wide range of energy-based Hamiltonian formulations in quantum-mechanical systems. One line of research that has been more robustly enabled is that of scrambling: complex systems (such as brains, black holes, and many-body quantum systems) are posited to be fast-scramblers, dissipating information quickly such that a local measurement is no longer possible [141]. Various SYK Hamiltonians and scrambling Hamiltonians [142] could be applied in the AdS/Brain model structure to formalize neural signaling as a quantum information scrambling problem.

*4.5. Neuronal Gauge Theories*

Neuronal gauge theories comprise another class of neuroscience physics approaches. One symmetry-based project models the brain's neural signaling operation on the basis of gauge invariance and global symmetry [143]. Symmetry is the property of physical systems looking the same from different points of view (whether a face, a cube, or the laws of nature), and symmetry breaking is phase transition. A gauge theory is a field theory in which the Lagrangian (state of a dynamic system) does not change (is invariant) under local gauge transformations (changes between possible gauges (levels or degrees of freedom) in a system). This neuronal gauge theory interprets the brain as a multiscalar system with a global symmetry, the invariant property of free energy minimization, that is broken and rebalanced. Neural signaling breaks the symmetry and gauge fields are applied to rebalance the invariant quantity (free energy). The gauge fields are part of the brain environment and apply continuous forces to act on the brain elements to produce local perturbations that counteract the effect of the local force stimulus as neural signals are dispatched in order to bring the system back into its resting state. The gauge field rebalancing mechanism coordinates the multiscalar tiers of the brain on the basis of conserving the gauge-invariant quantity, free energy minimization in this model, but could be otherwise.

Another neuronal gauge theory formulation [144] finds that the macroscopic brain obeys the same kind of energy–entropy balances when at rest as microscopic processes and likewise breaks the balance when performing physically and cognitively demanding tasks (according to connectome project whole-brain imaging data [145]). Non-equilibrium processes at the macroscale are studied with a dynamic Ising model showing how violations of the energy–entropy balance emerge from asymmetries in the interactions between neural elements. A related gauge-theoretic model explores energy–entropy trade-offs in the relation between the information content of brain states and neural energy [146]. Brain

states are modeled as the Shannon entropy content of parcels, and energy via the Boltzmann distribution, as the brain network seeks lower-energy stable states. The multiscalar energy–entropy model is applied to explain how signal propagation along the structural connectome of the brain may induce changes in the patterns of neural activity (again similarly tested with empirical connectome data).

### 4.6. Network Neuroscience

Network neuroscience is a (quantum) information network neurobiology program that takes a complex systems [20], network [147], physics [148], energy [146], and information-based view of neurobiology [149]. The program unites elements of the brain network-level view, energy, entropy (information), neural dynamics, structural–functional linkage, multi-scalar systems and renormalization, microscale–macroscale interactions, an energy economy view of the brain, and information signaling theory (encoding and compressibility), validated with empirical data from the human connectome project and other sources [145]. Recent findings of note are in the areas of renormalization, neural dynamics, neuronal gauge theories, and information-based encoding.

Overall, the brain is seen as an information problem that can be modeled with entropy and energy, with potential translation to quantum platforms. Renormalization techniques might sidestep the usual difficulties of integrating a multiscalar environment by modeling behavior as an information compression problem in which similar constraints impact all tiers [149]. Likewise, network architecture and connectivity are indicated as system-wide parameters that influence multiscalar behavior; for example, contributing to an oscillatory-based understanding of local and global neural dynamics [150].

Network neuroscience sees practical demonstration by other connectome teams using graph theory and differential geometry to study the spatiotemporal arrangements of neurons, synapses, axons, and dendrites. A quantum approach to connectome analysis calculates the eigenvectors of the human connectome with a graph Laplacian (Schrödinger wavefunction element) [151]. The resulting harmonic wave is used to examine neural fields as a basis for structure–function relationships in the human brain, implementing the Wilson–Cowan neural field theory with high-resolution MRI connectome graph data.

### 4.7. Random Tensors

Random tensors are a tensor network technology for the treatment of high-dimensional multiscalar systems—an advance on par with MERA tensor networks (computation of entangled quantum systems). Tensor networks are a structure for representing and manipulating many-body quantum states as the factorization of high-order tensors (tensors with a large number of indices) into a set of low-order tensors whose indices are summed to form a network defined by a certain pattern of contractions. Random tensors generalize random matrices ($2 \times 2$ matrix formulations) to three or more dimensions and have been tested for as many as five dimensions (rank-5 tensors) [152].

Random tensors provide another model (in addition to matrix mechanics) for the implementation of the AdS/Brain theory as a tensor field theory of neural signaling. Existing neural field theories could be instantiated (with three-state neurons [88]) as tensor field theories [153] on quantum platforms. Likewise, the four dimensions (network–neuron–synapse–molecule) of the AdS/Brain theory could be indexed with rank-4 random tensors, modeling the quiescent-to-firing signal as a matrix(2d)-to-tensor(3+d) phase transition (planar-to-melonic (high-dimensional) graph representation).

These kinds of tensor field theories and melonic graphs of neural signaling have the dimensionality needed to instantiate synaptic integration research findings, extending the sophistication of traditional computational neuroscience compartmental models. Spine density gradients are known to be important in shaping dendritic response, and decreasing spine density improves thresholded signal pooling (certain neurons pool the outputs of many separately thresholded dendrites) [154]. Nonlinear models have also been used to study the postsynaptic density and dendritic shape as elliptical spheroids, finding that the

curvature of dendritic geometry gives rise to pseudo-harmonic functions that can be used to predict dendrite concentrations and their potential role in signal processing [155].

Differential geometry implementations bring a new resolution to the study of mitochondrial membrane architecture, whose metabolic impairment may contribute to neurodegeneration [156]. Traditional ways of modeling mitochondria (ATP and heat) are insufficient as their idealized geometries distort metabolic flux. Applying differential geometry to empirical TEM tomography data, however, allows a more robust analytic model based on Gaussian curvature, surface area, volume, and membrane motifs, all of which are related to the metabolic output of the mitochondria and require a multi-dimensional approach. Such differential geometry methods might inflect into practical applications treating mitochondrial bioenergetic stress response [157].

## 5. Discussion

The study of neuroscience is necessarily migrating to quantum information science platforms, as quantum computing may become the computational vernacular of the day. However, simply reinstantiating research programs with quantum information science methods is not likely to solve neuroscience problems as expediently as also incorporating the theoretical understanding available in foundational physics discovery. Hence, the emerging field of quantum information science-driven quantum neurobiology is outlined in the three levels of its activity in wavefunction analysis, neural dynamics, and neuroscience physics. These areas include the clinically motivated investigation of wave imaging, protein folding, and genomics, the study of neural dynamical systems with superpositioned data, quantum probability, and neural field theories, and the neuroscience physics interpretation of physics findings. A novel quantum theory of neural signaling is proposed—the AdS/Brain theory, as the first instance of a multi-tier AdS/CFT correspondence model of successive levels of bulk–boundary relationships between network, neuron, synapse, and molecular levels in the brain. Quantum solutions to key neural signaling challenges, the synaptic integration of thousands of incoming signals and electrical–chemical signal transduction, are proposed in several modalities.

There are many potential risks and limitations to quantum neurobiology. It may be too early for quantum technologies since technical breakthroughs in quantum error correction are needed to progress from NISQ (noisy intermediate-scale quantum) devices to fully FTQC (fault-tolerant quantum computing) [158]. Modeling the complexities of the brain may not be a near-term application even if quantum methods proceed. The challenge of obtaining empirical data (due to both technical and privacy-related reasons) constrains the ability to develop accurate neurobiological models. However, despite these limitations, quantum neurobiology is extending the study of neuropathological disease.

Future directions in quantum neurobiology could explore topics linking physiological building blocks to higher-order cognitive behavior as structural-functional relationships are uncovered in genomics, connectomics, and cortical recording studies. Quantum information science and quantum measurement theories might be deployed to understand not only neurobiological behavior but also psychological information processing [159] and human decision-making [160], including via quantum BCI implementation.

Quantum neurobiological approaches could seek to interrelate three domains: scientific theories of time, temporal modes of cognition, and the underlying time morphology of biological processes. Physics findings related to time include time entanglement (temporal correlations have a different structure than spatial correlations [161]), the Floquet model [134], time symmetry breaking [162], time evolution [141], and quantum walks [136]. In the cognitive domain, Kantian neuroscience shows empirically how the spontaneity of cognition is demonstrated by the constitutive role of the brain in processing incoming sensory input [163] and correlates neurobiology and philosophy [164]. These approaches could be further linked to the temporality of biological processes (cellular lifecycles, oscillatory patterns, and circadian rhythms) [165].

The accelerated pace of the technology-driven "big data" era has sponsored the development of a computational informatics field as a complement to traditional academic disciplines in many areas of the arts and sciences (ranging from digital humanities to computational astronomy). Likewise, "quantum studies" fields are emerging as an accompaniment, pursuing quantum approaches to the underlying questions in the discipline and enabling a new class of more precise problem-solving, thinking, and discovery, with a standard slate of quantum information science methods [166]. Quantum neurobiology is in the early stages of development but could potentially have an extremely transformative impact on the ability to elaborate the intricacies of the human brain and better protect it from disease and decline. A better understanding of the neurobiological role of quantum effects (or quantum-analog effects) may also shed new light on fundamental interpretative questions in quantum physics.

**Author Contributions:** Conceptualization and writing; M.S.; review and suggested additions; R.P.d.S. and F.W. All authors have read and agreed to the published version of the manuscript.

**Funding:** This research received no external funding.

**Conflicts of Interest:** The authors declare no conflict of interest.

## Glossary

| | |
|---|---|
| AdS/Brain | Multiscalar neuroscience interpretation of the AdS/CFT correspondence |
| AdS/CFT correspondence (anti-de Sitter space conformal field theory) | Theory positing that a physical system with a bulk volume can be described by a boundary theory in one less dimension |
| AlphaFold | Protein folding predictor based on system-level attention to spatial constraints (from DeepMind/Google) |
| Biological physics | Study of living processes through the application of physical principles |
| Bosonic codes | Self-contained photonic system for quantum error correction (e.g. harmonic oscillator) |
| Chaotic dynamics | Dynamical regimes of ballistic spread followed by saturation |
| Filamentary dynamics | Role of neurofilaments (neuron-specific proteins) in axonal and synaptic signaling |
| GKP bosonic codes (Gottesman, Kitaev, Preskill) | Quantum error correction method by reorienting the position and momentum of a molecule with known symmetric rotations |
| Hamiltonian | (Quantum mechanics) operator corresponding to the total energy of a system |
| Hamming distance | Sum of positional mismatches of two bit strings |
| Hopf bifurcation | System critical point at which a periodic orbit appears or disappears per a local change in stability |
| Information biology | Study of information processing activities performed by biosystems |
| Information scrambling | Rapid spread of information in a quantum system prohibiting local measurement |
| Josephson junction | Device consisting of two or more superconductors coupled by a link that conducts electrons |
| Laplacian | (Schrödinger equation) operator representing the flux density of the gradient flow of a function |
| Matrix | Array of numbers arranged in rows and columns used to study physical phenomena (probability distribution) |
| Melonic diagram | (Melon-shaped) graph expression of a high-dimensional system |
| MERA (multiscale entanglement renormalization ansatz) tensor networks | Entangled quantum systems model |
| Molecular codes | Quantum error correction by performing rotations on asymmetric rigid bodies in free space |
| Nanoparticle neuroscience | Nanoparticles (100 nm objects) that cross the blood-brain barrier to perform an intervention |
| Neurobiology | Field investigating the form and function of the nervous system (neurons, glia, axons, and dendrites) |
| Neurofilament | Neuron-specific protein implicated in neuronal cytoskeletal structure and signaling |
| Neuromorphic computation | Electronic computation inspired by neural systems and spike thresholding |
| Neuropeptides | Small chains of amino acids (chemicals) synthesized and released by neurons |
| Neuroscience | Study of the structure and function of the nervous system and brain |
| Neuroscience physics | Neuroscience interpretation of foundational physics findings |
| Path integral | Approach of summing over all possible paths in a system |
| Protein folding problem | Predicting a protein's final 3D structure from the underlying sequence of amino acids |
| Quantum biology | Study of how quantum properties may play a governing role in biological functions |
| Quantum computing | Use of engineered quantum systems (with atoms, ions, photons) to perform computation |
| Quantum information biology | Study of biological systems with quantum information methods |
| Quantum internet | Information transmitted with quantum effects (entanglement), using quantum cryptography |
| Quantum machine learning | Machine learning applied in a quantum environment |
| Quantum memory (QRAM) | Quantum-mechanical computer memory, storing information with greater scalability as quantum states in superposition (vs classical binary states) |
| Quantum nanoscience | Study of nanostructured systems that incorporate and exploit quantum effects |
| Quantum neurobiology | Discipline within quantum biology and biological physics that studies potential quantum effects in the brain and applies quantum information science methods to neurobiological questions |

| | |
|---|---|
| Quantum physics | Description of particles making up all matter including living organisms |
| Quantum probability | Quantum mechanical rules for assigning probabilities |
| Quantum walk | Quantum version of classical random walk based on coin-flip operator and lattice-graph propagation |
| Qutrit | Three-level quantum state, simultaneously in 0, 1, 2 until collapsed in a measurement (vs two-state qubit) |
| Random tensors | Generalization of random matrices ($2 \times 2$ matrix formulations) to 3+ dimensions |
| Renormalization | The ability to view a system at multiple scales by collapsing degrees of freedom (parameters) |
| Spike-activated neural networks (SNNs) | Bio-inspired neuromorphic computation based on thresholded activation |
| Superdeterminism | Interpretation that quantum effects are the result of hidden variables (vs indeterminism) |
| Superpositioned data | Quantum information representation of all possible system states simultaneously |
| Tensor field theories | Local field theories whose fields transform as a tensor under a global or local symmetry group |
| Tensor networks | Structure for manipulating high-dimensional data (many-body quantum states) as the factorization of high-order tensors (many indices) into low-order tensors whose indices are summed to form a contracted network |
| Transcription factors | Proteins regulating gene expression by attaching themselves to DNA |

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
