# Peer review of "Quantum Neurobiology"

_quantumrep, doi:10.3390/quantum4010008_

Round 1

Reviewer 1 Report

The manuscript is currently in good format, clearer, having good flow of content and thought, and more readable even by the clinical neuroscientists.

Reviewer 2 Report

The revised version addressed all the major and minor issues and improved significantly. I think such a review would be highly interesting for the people who are working in the biological fields, precisely on neuron and artificial brain-building like me. I shall keep a copy of this paper after getting published with me. I wouldsuggest the editor publish this review in your journal. 

Reviewer 3 Report

The authors modified the manuscript according to my comments and I believe that it can be published in the present form.

This manuscript is a resubmission of an earlier submission. The following is a list of the peer review reports and author responses from that submission.

Round 1

Reviewer 1 Report

The manuscript by M. Swan, R. P. dos Santos, and F. Witte addresses the applications of quantum information methodology to multiscale problems of a human brain. This emerging field is comprehensively reviewed and a novel approach, AdS/Brain, is introduced (this acronym should be defined when first appeared.) I believe that this paper would be interesting for a broad audience, but before I recommend it for publication, a major revision is necessary.

In some sense, even the title “Quantum neurobiology” is somehow misleading. A reader would expect a discussion similar to the definition in the paper by S. Tarlacia and M. Pregnolato, Quantum neurophysics: From non-living matter to quantum neurobiology and psychopathology,  International Journal of Psychophysiology 103, 161-173 (2016), Quantum neurobiology is a concept to which we are not yet fully accustomed to: it refers to a narrow field of the operation of quantum physics in the nervous system such as the emergence of higher cognitive functions like consciousness, memory, internal experiences, and the processes of choice and decision-making which are products of the warm-wet-noisy brain. According to quantum neurobiology, quantum physics is involved in biological processes, and consciousness, memory, internal experiences, and the processes of choice and decision-making, which are the products of the warm-wet-noisy brain, may be the result of the operations of quantum physics. According to this, and in contrast to the classical view, the information processing units are not the nerve cells but smaller quantum physical processes inside the cells and forming connections between them. However, the authors of the present manuscript state, The potential role of quantum effects in the brain is a controversial topic. Many scientists note that they do not endorse this idea (the quantum cognition hypothesis [10, 11]). Some argue that the brain is simply too large and too warm to exhibit quantum effects [12], and has short decoherence timescales [13]. Quantum neurobiology is inspired by the mathematical structure of quantum mechanics, but not the conjecture that there is something quantum-like taking place in the brain [14, 15]. Although I completely agree with the last statement, this definition of quantum neurobiology deviates from the common perception. It should be also noted that the authors mentioned only the old idea of Penrose and Hameroff of the quantum computation in the brain based on microtubules but ignored more recent proposal by Matthew Fischer (Quantum Cognition: The possibility of processing with nuclear spins in the brain, M.P.A. Fisher, Annals of Physics 362, 593-602 (2015)) based on nuclear spins with much longer decoherence time. Nevertheless, it should be stated in the very beginning and be clear even from the title that quantum processes in the brain are not the topic of the present paper.

The part related to the system evolution should be also revised. Authors write, Central to quantum modeling is the need for sophisticated dynamics frameworks to evolve systems that extend beyond the traditional approaches of Schrödinger and Heisenberg dynamics [93]. The Heisenberg equation of motion is only a general approximation of movement and does not include temperature (thermality being an important attribute in biological, superconducting, and black hole systems) [94]. The Schrödinger wavefunction is limited to describing pure quantum states as opposed to mixed states (combinations of states). The two main approaches to next-generation quantum dynamics are quantum master equations and ladder operators. In this, they refer to the textbook applications of Schrödinger and Heisenberg approaches to model systems with no environmental degrees of freedom that would provide the temperature. Quantum dynamics in the presence of environment is discussed within the so-called theory of open quantum systems with thousands of papers published on that subject. In particular, both theoretical papers already published in this Special Issue use some versions of this theory. Two extensively cited books, Quantum Dissipative Systems by U. Weiss and Theory of Open Quantum Systems by H.-P. Breuer and F. Petruccione, describe various approaches with quantum master equations being only one of them. Referral to ladder operators as a second next-generation approach is wrong. Ladder operators is just a technique. Equations of motion for them are the Heisenberg equations. Moreover, for a harmonic oscillator, the ladder operators are superpositions of the coordinate and momentum operators. For many situations, it is more convenient to treat the system-environment interaction in terms of the ladder operators, but it can be done for the operators of physical measurables as well within the same approach.  

The sentence in lines 299-301 does not make sense. It says, The result is a master equation written as a neural Hamiltonian, as an evolution operator that counts all the possible state transitions in and out of the system states, in spatial and temporal dimensions, and is equal to the total energy of the system. A master equation cannot be written as a Hamiltonian, which is indeed the operator of the total energy, they can be derived from it. Hamiltonian is not the evolution operator which is an exponent with the argument proportional to the Hamiltonian.

In summary, the revised manuscript should distance itself from the quantum processes and the parts related to quantum evolution should be rewritten or even eliminated.

Reviewer 2 Report

The authors provide a comprehensive review on a new generation of bio-inspired quantum technologies and physics-inspired theory of neural signaling and information storage and transfer in the emerging field of Quantum Neurobiology. Since human brain is regarded as the most complex anatomical structure (I prefer not to use organ for the brain, because in my opinion the brain cannot be transplanted – brain with quantum and consciousness) and thus rightly stated in the manuscript as having a multiscalar problem; hence multidisciplinary approaches or knowledge-fusion to study the brain is required. This manuscript which contains recent citations is viewed as an important scientific introduction in the emerging field of Quantum Neurobiology for non-computer, non-mathematics or non-physicist scientists such as fundamental structural-functional neuroscientists, academic neuro-philosophers and clinicians. In future, perhaps the authors should also review articles related to ‘Quantum Plasma Brain Dynamics’ (a fusion of Quantum with Plasma or Thermodynamics Physics for the brain)(which lacking in this review manuscript), where there are plenty of extracellular plasma (ionic gases) being produced by: a) astrocytes-aquaporin (wet vs dry brain area), and b) light (Quantum Field) interaction with the electrons (Electromagnetic Field) producing nanoplasmonic and coherence thermal waves. Thus, brain can be viewed as having both partially Mechanical (Classical/Newtonian and Quantum Mechanics) and partially Thermodynamical Systems.

Overall, it is an excellent review that needs few corrections as my suggestion to improve it:

  1. Since the title is ‘Quantum Neurobiology’ and has many ‘Quantum’ word stated in the manuscript. Perhaps, it is better to have a brief explanation on the meaning of this word (Quantum) at introduction. By doing so, other neuro-field scientists and neuro-clinicians can grasp the content easier. Having a brain figure at introduction to depict this (or a figure of Quantum brain concept with those 3 tiers of application) may certainly improve readers’ understanding
  2. The manuscript should be identified as a review article (not as a research article)

Reviewer 3 Report

The article on "Quantum Neurobiology " by Swan et al is aiming to improve the understanding of neuropathologies of disease, real-time biological data processing, neuronanorobotic health monitoring and the understanding behind the overall information processing. However, This article is written in a mini-review style, where the fundamental research work is absent, which is the basic criteria for writing a manuscript in the 'Article' format. However, like 'Mini Review' the all the points are even not explained properly and to illustrate the text, some figures are essentials. Apart from the technical issue, some scientific concerns should be addressed. The quantum effect in neurons is considered fundamentally for the nine orders-of-magnitude scales in length from the atomic and cellular level to local brain networks, which is never demonstrated experimentally. By varying the length only on the same materials, the quantum effect can not be observed. A neuron process the information on various time domains [J Neurophysiol (doi:10.1152/jn.00478.2020), J. Integr. Neurosci. (doi:10.31083/j.jin2004082], which basically promotes the optical vortex formation ( Symmetry 2021, 13, 821. https://doi.org/10.3390/sym13050821). These quantum optics experiments demonstrate the different time domains of information processing by different parts inside the neuron. This look may be very new, but the way the manuscript is written can not be considered for publishing it as an 'Article' in the 'quantum report' journal.